# Origin and Genealogy of Rare mtDNA Haplotypes Detected in the Serbian Population

**DOI:** 10.3390/genes16010106

**Published:** 2025-01-20

**Authors:** Slobodan Davidović, Jelena M. Aleksić, Marija Tanasković, Pavle Erić, Milena Stevanović, Nataša Kovačević-Grujičić

**Affiliations:** 1Group for Human Molecular Genetics, Institute of Molecular Genetics and Genetic Engineering, University of Belgrade, Vojvode Stepe 444a, 11042 Belgrade, Serbia; milena.stevanovic@imgge.bg.ac.rs (M.S.); natasa.kovacevic@imgge.bg.ac.rs (N.K.-G.); 2Department of Genetics of Populations and Ecogenotoxicology, Institute for Biological Research “Siniša Stanković”—National Institute of the Republic of Serbia, University of Belgrade, Bulevar Despota Stefana 142, 11060 Belgrade, Serbia; marija.tanaskovic@ibiss.bg.ac.rs (M.T.); pavle.eric@ibiss.bg.ac.rs (P.E.); 3Group for Cardiovascular Physiology, Institute for Medical Research—National Institute of the Republic of Serbia, University of Belgrade, Dr. Subotića 4, P.O. Box 39, 11129 Belgrade, Serbia; jelena.aleksic@imi.bg.ac.rs; 4Serbian Academy of Sciences and Arts, Kneza Mihaila 35, 11000 Belgrade, Serbia

**Keywords:** human populations, Balkan Peninsula, mtDNA, genetic diversity, high-resolution phylogeny, human migrations

## Abstract

**Background**: The Balkan Peninsula has served as an important migration corridor between Asia Minor and Europe throughout humankind’s history and a refugium during the Last Glacial Maximum. Past migrations such as the Neolithic expansion, Bronze Age migrations, and the settlement of Slavic tribes in the Early Middle Ages, are well known for their impact on shaping the genetic pool of contemporary Balkan populations. They have contributed to the high genetic diversity of the region, especially in mitochondrial DNA (mtDNA) lineages. Serbia, located in the heart of the Balkans, reflects this complex history in a broad spectrum of mtDNA subhaplogroups. **Methods**: To explore genetic diversity in Serbia and the wider Balkan region, we analyzed rare mtDNA subclades—R0a, N1a, N1b, I5, W, and X2—using publicly available data. Our dataset included already published sequences from 3499 HVS-I/HVS-II and 1426 complete mitogenomes belonging to West Eurasian and African populations, containing both contemporary and archaeological samples. We assessed the parameters of genetic diversity found in different subclades across the studied regions and constructed detailed phylogeographic trees and haplotype networks to determine phylogenetic relationships. **Results**: Our analyses revealed the observable geographic structure and identified novel mtDNA subclades, some of which may have originated in the Balkan Peninsula (e.g., R0a1a5, I5a1, W1c2, W3b2, and X2n). **Conclusions**: The geographic distribution of rare subclades often reveals patterns of past population movements, routes, and gene flows. By tracing the origin and diversity of these subclades, our study provided new insights into the impact of historical migrations on the maternal gene pool of Serbia and the wider Balkan region, contributing to our understanding of the complex genetic history of this important European crossroads.

## 1. Introduction

The Balkan Peninsula has played an important role in the history of the human population in Eurasia, serving as an important migration corridor from the Palaeolithic era to the present-day [1,2,3], not only as a bridge between Asia Minor and Europe [4,5] but also as one of the glacial refugia during the Last Glacial Maximum [6,7,8]. The region’s turbulent history suggests that several different migration events shaped the gene pool of contemporary Balkan populations [9,10]. Agriculture arrived in Europe from the Middle East during the Neolithic expansion through Asia Minor and the Balkans, as documented with archaeological remains from some of the oldest Neolithic farmer societies found in Europe (e.g., Starčevo and Vinča cultures) [11,12]. Analysis of ancient DNA from the Neolithic populations of the Balkan Peninsula has revealed a significant admixture between autochthonous hunter-gatherer populations and newly arrived farmer populations from northwestern Anatolia [4]. Additionally, several studies indicate that Neolithic expansion took two distinct routes through the Balkans: one following the Danube River, reaching central Europe, and another along the Mediterranean route, reaching the Iberian Peninsula [4,12].

The second historical event that significantly shaped the genetic pool of Balkan populations occurred during the Early Bronze Age, when steppe tribes of the Yamnaya culture initiated a chain of migration events followed by their admixture with the indigenous groups, which impacted not only southeast but also Central Europe [13,14,15,16,17]. During the Migration Period in the Early Middle Ages (fourth to ninth century A.D.), the genetic landscape of human populations further changed in southeast, Central, and East Europe, significantly influencing the genetic variability of modern European populations [18]. Analysis of segments identical by descent revealed that these populations share a substantial number of common ancestors dating back to this period [18]. In the Balkan Peninsula, the most significant event of this period was the large-scale settlement of Slavic tribes. This event not only defined the region’s cultural trajectory, as most Balkan populations today speak one of the South Slavic languages [19], but was also one of the largest permanent demographic changes in Europe during the entire Migration Period [20]. Hence, the Balkans’ turbulent history of migration and colonization processes led to high genetic diversity observed in the Balkan human population [9] and greater haplotype diversity compared to Northern Europe [21] as well as several micro-differentiation processes in isolated populations [22]. All these results highlight the importance of southern Europe, particularly the Balkan region, as a crucial reservoir of genetic variation for European populations [5,9,20,22].

The Republic of Serbia is positioned in the central part of the Balkan Peninsula. Studies of the variability and distribution of mitochondrial DNA (mtDNA) (sub)haplogroups reveal that Serbian population occupies a space between the two groups of South Slavic populations inhabiting the eastern (Bulgaria and North Macedonia) and western (Slovenia, Croatia, Bosnia and Herzegovina) part of the Balkan Peninsula [9,19] and correlates with its geographic position. Studies based on Y-chromosome haplogroups show similar relations [10]. Furthermore, the Serbian population exhibits high heterogeneity of mtDNA (sub)haplogroups [9], indicating that various evolutionary processes shaped its maternal genetic landscape as well [9,23,24]. Phylogeographic analyses of super haplogroup U lineages found in the contemporary Serbian population further corroborated that its gene pool was shaped by the migrations from the Neolithic, Bronze Age, and Migration Period [5,23], while analyses of several rare mtDNA lineages (H6a2b, L2a1k, U1a1c2, U4c1b1, U5b3j, and K1a4l) found in this population imply that they have possibly evolved in the Balkan Peninsula [9,23]. All these findings suggest that the Serbian population could be representative for the genetic diversity studies for both South Slavic and Balkan populations.

Analyzing genetic variability within specific mtDNA subclades and their geographic distribution can help identify the region where these subclades may have originated. Determining the source population and the geographical regions where specific subclades have evolved is important for tracing the direction of the migrations and gene flow between various populations. For this type of analysis, the rare subclades can provide more information than the frequently found ones, among different populations [25]. The value of investigating the genealogy of low-frequency haplogroups with relic distribution to trace human migration routes was previously already demonstrated for R0a, N1a1a, and X haplogroups [25,26,27,28].

Although haplogroup R0a is rare in Europe, its frequency in the Arabian Peninsula can be as high as 17% [29,30]. According to Gandini et al. [25], the dispersion of R0a lineages from the Levantine refugium to Europe most likely occurred during the Late Glacial/Early Postglacial period. Similarly, subclade N1b is rare in Europe but can be found with almost doubled frequencies in the Middle East [26]. Most likely, N1b carriers arrived in Europe during the Late Glacial, allowing enough time for certain lineages, like N1b2, to evolve within Europe [26]. Subclade N1a and some of the subclades of haplogroup W, together with T2, K, J, HV, V, and X were part of the so-called mitochondrial “Neolithic package”, characteristic of early Neolithic farmers from Central Europe [31]. During the Late Neolithic and Bronze Age, haplogroup N1a almost disappeared in European populations and new mtDNA haplogroups I, T1, U2, U4, U5a, W, and subtypes of H replaced early Neolithic haplogroups [31,32,33]. To identify the centers of diversity for these rare subclades in Europe and reconstruct the migration routes, we have analyzed the genetic diversity of selected rare subclades found within the different regions of West Eurasia and Africa.

To assess the geographical origin of the haplotypes found in the Serbian population belonging to subclades rare in European populations, such as R0a, N1a, N1b, I5, W, and X2, we conducted a detailed analysis of available data for mtDNA HVS-I and HVS-II sequences, as well as complete mitogenomes from West Eurasian and African populations. Our aim was to evaluate the impact of different migrations on shaping the contemporary gene pool of the Serbian population and the populations in the region from a maternal perspective.

## 2. Materials and Methods

### 2.1. Sample

MtDNA haplotypes belonging to (sub)haplogroups R0a, N1a, N1b, I5, W, and X2 were collected from available published data (Appendix A). A total of 3499 HVS-I/HVS-II haplotypes and 1426 complete mitogenomes, classified into the mentioned rare (sub)haplogroups, were collected for the analyses. Out of the total analyzed samples, 111 sequences originated from archaeological remains. Haplotypes were defined by the variations in HVS-I (nucleotide positions (nps) 16,024–16,400) and HVS-II (nps 72–340) sequences of the control region or the complete mitogenomes where available. Haplotypes were reconstructed using the HaploSearch program [34] for the samples whose haplotypes were not provided in the publications and those with only available sequences. For samples that lacked haplogroup classification in the original publications we have performed haplogrouping using the Haplogrep 2 and Haplogrep 3 software [35,36]. In addition, complete mitogenome sequences with haplogroup classification were extracted from Ian Logan’s website (http://www.ianlogan.co.uk/sequences_by_group/haplogroup_select.htm, accessed on 18 June 2024) which is regularly updated.

### 2.2. Statistical Analysis, Phylogeography, and Phylogeny

The frequency distribution of mtDNA (sub)haplogroups, determined by both HVS-I and HVS-II sequences, were extracted from the published data, pooled together, and assessed for each geographic region. Choropleth maps were generated using the R statistical programming language (version 4.1.2) [37]. Ethnic groups with distributions across national state borders were not visualized on the map. Geospatial polygons that represent countries and regional boundaries were obtained from the “rnaturalearth” package (version 0.1.0) [38] and merged to represent the geographical regions from the published data using the “sp” R package (version 1.5.1) [39,40]. Choropleth maps were plotted using the “ggplot2” R package (version 3.3.6) [41].

Genetic diversity in different geographical regions (Balkan Peninsula, Apennine Peninsula, Iberian Peninsula, Eastern Europe, Central Europe, Western Europe, Northern Europe, Near East, Asia—without Near East and Africa) was assessed by measuring parameters of genetic diversity based on the variability of HVS-I/HVS-II sequences of the haplotypes belonging to the (sub)haplogroups R0a, N1a, N1b, I5, W, and X2that originated from specified geographical regions. Calculations were performed using Arlequin ver. 3.5.2.2 [42].

Phylogeographic analysis based on the variability of HVS-I and HVS-II sequences was performed by constructing a median-joining network using Network ver. 10.2.0.0 (http://www.fluxusengineering.com/network_terms.htm, accessed on 9 May 2024). Different weights were assigned to observed substitutions based on their evolutionary rates [43]. Postprocessing MP calculations were performed, and the resulting networks were manually assembled for visualization. Point indels located between nps 16,180–16,193 and 303–315 were not used for these analyses.

Phylogeny reconstruction was performed for the available complete mitogenomes classified into the appropriate (sub)haplogroups using mtPhyl v4.015 software (https://sites.google.com/site/mtphyl/home, accessed on 18 January 2024). The program was manually updated to follow the mtDNA phylogeny available in the PhyloTree build 17, and the following literature [9,22,23,44,45,46,47,48,49,50]. New subclades were defined when they comprised at least two different mitogenomes with at least one shared mutation not characterized as a hotspot [32]. The length variations at nps 303–315, 522–524, 573–576, and 16,180–16,193, the polymorphism at np 16,519, and A–C transversions at nps 16,182 and 16,183 were excluded from phylogenetic analysis.

## 3. Results

### 3.1. Subhaplogroup R0a

The R0a subclade is the most frequent in Yemenis (17.33%) and Bedouins (14.77%) (Figure 1 and Appendix A). We observed the highest values for the analyzed genetic diversity parameters in the Near East, the Apennine, and Balkan Peninsulas (Appendix A).

A haplotype network for the subhaplogroup R0a was constructed using 217 HVS-I/HVS-II haplotypes detected in 613 individuals (Figure 2). In the Serbian population, subclade R0a is represented by a single haplotype found in sample 98_Sb [24]. This haplotype differs from the founder HVS-I/HVS-II haplotype, which is predominantly detected in the Balkan and Arabian Peninsulas, by a transition at np 16,248 (Figure 2, Appendix A).

Phylogeny based on the 71 available complete mitogenomes showed that the sample 98_Sb, together with two haplotypes from Bulgaria, belongs to the newly identified branch, R0a1a5, defined by the transitions at the nps 11,914, 12,189, and 1458 (Appendix A). This newly defined branch is estimated to have evolved 1.54–1.8 thousand years ago (kya) (Appendix A).

### 3.2. Subhaplogroup N1a

The N1a subhaplogroup is most prevalent in Kuwait, while the highest frequency in Europe can be detected in Estonia (Figure 1, Appendix A). As a European region, the Balkan Peninsula shows the highest occurrence of N1a (Figure 1, Appendix A). The highest values of its genetic diversity parameters were detected in Asia and the Near East (Appendix A).

A haplotype network representing the N1a subhaplogroup was constructed using 172 HVS-I/HVS-II haplotypes detected in 327 individuals (Figure 3). In the Serbian population, the N1a subclade is represented by eight different HVS-I/HVS-II haplotypes (Figure 3 and Appendix A). Two of the haplotypes (sample 154_Sb [24] and Ser_04 [51]) were characterized by the 16,147A transversion and accordingly classified into the European branch of subhaplogroup N1a [52]. On the other hand, four haplotypes (33_Sb [24], VP87 [53], Ser_03 [51] and 232_Sb [24]), are characterized by the 16,147G transversion and classified into the African/South Asian branch of subhaplogroup N1a [52] (Figure 3, Appendix A). Interestingly, a new branch with a back mutation at np 16,147 is identified and samples 242_Sb [24] and Nish8 [5] belong to this branch (Figure 3).

Phylogeny reconstruction based on 81 complete mitogenomes classified the 154_Sb haplotype into the N1a1a1a1 subclade that emerged around 4.45–4.38 kya (Appendix A). This haplotype is identical to three haplotypes from Eastern Europe (Russian and Estonian) and one from the Balkan Peninsula (Bulgarian), while other N1a1a1a1 mitogenomes originate from various populations of Eastern and Central Europe, the Balkan Peninsula, and some Central Asian populations (Appendix A). The 242_Sb haplotype and the mitogenome from Russia form a new branch N1a1a1a1b, dated to 2.71 kya and defined by the transition at np 11,167 (Appendix A). The Serbian haplotype also has back mutations at nps 16,172 and 16,147, differentiating it from the haplotype found in the Russian individual (Appendix A).

Complete mitogenomes of samples 33_Sb and 232_Sb belong to the N1a1a2 branch (1.97–2.31 kya) along with one haplotype from Russia and one haplotype of unknown origin (Appendix A).

### 3.3. Subhaplogroup N1b

Subhaplogroup N1b is mostly present in the Near East, with a frequency of up to 5.67% in Lebanon (Figure 1 and Appendix A). It is scarce in Europe, exhibiting the highest frequencies in Sicily (3.85%) and Romania (2.10%) (Figure 1, Appendix A). Although the highest number of N1b haplotype carriers is found in the Near East, the highest haplotype and nucleotide diversity values were observed for the Balkan Peninsula (Appendix A).

A haplotype network for the subhaplogroup N1b was constructed using 154 HVS-I/HVS-II haplotypes detected in 396 individuals (Figure 4). Six HVS-I/HVS-II haplotypes belonging to subhaplogroup N1b were detected in the Serbian population (Figure 4). The haplotype found in sample 143_Sb [9] was identical to four haplotypes from the Balkan and the Apennine Peninsulas (Figure 4, Appendix A). Together with another Serbian haplotype Nish9 [5], it is positioned in the branch defined by 16,716A transversion that includes haplotypes originating from the Near East (Ashkenazi Jews), Central Europe, the Balkan, and the Apennine Peninsulas (Figure 4, Appendix A).

Three different HVS-I/HVS-II haplotypes found in the Serbian population (samples 51_Sb [9], Studenica20 [5], VP88 [53], and SS22 [47]) were positioned within the branch defined by the transition at the np 152 and as a majority of haplotypes in N1b network, they have 16,176G transversion (Figure 4). The haplotype detected in sample Nish33 [5] is identical to three haplotypes found in the Caucasus region in Armenia belonging to the branch differentiated from the ancestral haplotype by the transition at the np 185 (Figure 4, Appendix A).

Phylogeny reconstruction based on 109 complete mitogenomes, with two complete Serbian mitogenomes, allowed us to identify the new subclade N1b1a9, defined by the transition at np 14,323 and dated to 6.3–9.4 kya. This subclade groups haplotype detected in individual 51_Sb and the haplotypes from Sardinia (Appendix A). Within this novel subclade, two branches could be identified: N1b1a9a defined by transitions at nps 146, 152, 14,097, and 16,244, and N1b1a9b defined by transition at np 9064. The haplotype detected in individual 143_Sb is positioned in the N1b1a7′8 paragroup, defined by the transition at fast mutating np 195 (Appendix A).

### 3.4. Subhaplogroup I5

Subhaplogroup I5 is the second most frequent I subclade [54]. Worldwide, it is the most frequent in the northeastern part of Poland (2.74%) and the Greek population of Cyprus (2.2%) (Figure 1, Appendix A). The greatest number of I5 subhaplogroup carriers was detected in the Near East, the Balkan, and the Apennine Peninsulas (Appendix A), with the latter two regions exhibiting the highest frequencies of this subhaplogroup (Appendix A). The highest value for HD parameter was detected in the Balkan Peninsula, where high values of all other genetic diversity parameters were also observed (Appendix A). In the Serbian population, subhaplogroup I5 is represented by two haplotypes detected in samples 173_Sb and 256_Sb [24].

A haplotype network representing the subhaplogroup I5 was constructed using 72 HVS-I/HVS-II haplotypes detected in 168 individuals (Figure 5). The haplotype found in sample 173_Sb belongs to the branch defined by the 16,391 transition from the ancestral HVS-I/HVS-II haplotype (Figure 5). This branch comprises haplotypes found predominantly in the Balkan Peninsula (five haplotypes) as well as the Apennine Peninsula (one haplotype), along with a haplotype detected in one individual of Jewish origin from Morocco (Appendix A, Figure 5). The HVS-I/HVS-II haplotype of sample 256_Sb, together with two haplotypes from Iran and one from France, is located in the branch defined by the back mutation at the np 199 (Appendix A, Figure 5).

Phylogeny reconstruction based on 92 complete mitogenomes allowed us to classify the Serbian haplotypes in greater detail. The haplotype detected in sample 173_Sb was classified into the I5a1 subhaplogroup (Appendix A). This subhaplogroup, which evolved between 9.12 and 14.07 kya (Appendix A), contains haplotypes found mostly in European populations. One haplotype found in the sample from late 10th century CE Hungary (Anc10, [55]) is classified as I5a1a (Appendix A). A haplotype found in the individual 256_Sb and four haplotypes from Central Europe form a new subclade, I5d (12.61–14.75 kya), defined by transitions at nps 4532, 9156, 13,368, and 16,354 (Appendix A). Within this subclade, a new branch I5d1 (2.63–8.27 kya) is defined by a transition at np 7211 containing Serbian, German, and Polish haplotypes (Appendix A). Interestingly, in the HVS-I/HVS-II network (Figure 5), back mutation at the np 16,148 distances the 256_Sb haplotype from the haplotypes it is phylogenetically expected to be closer to according to the complete mitogenomes phylogeny (Appendix A). This disparity in genealogical relationships arises from the higher resolution of phylogenies based on complete mitogenomes, along with differences in sample size. The sample size is larger for the HVS-I/HVS-II-based haplotypes, which can offer varying perspectives on their genealogical relationships.

### 3.5. Haplogroup W

Haplogroup W is a rare haplogroup with the highest frequencies in the Georgian (9.42%) and Kurdish populations (6.67%) (Appendix A). In Europe, it is most frequent in the Albanian (5.42%) and Finnish (5.36%) populations (Appendix A, Figure 6). The highest numbers of haplogroup W carriers were found in Central Europe, the Near East, Northern Europe, and the Balkan Peninsula (Appendix A), and accordingly, the highest haplogroup frequencies were recorded in the Caucasus region of the Near East (5.06%) and Central Europe (2.96%) (Appendix A). High values of the genetic diversity parameters detected in the Balkan and the Apennine Peninsulas and Central Europe suggest that these regions could represent modern centers of diversity for haplogroup W (Appendix A).

A haplotype network representing the haplogroup W was constructed using 480 HVS-I/HVS-II haplotypes detected in 1026 individuals (Figure 7). Ten haplotypes classified into haplogroup W were detected among 20 individuals from the Serbian population (Figure 7, Appendix A). Detected haplotypes were classified into the following subclades: W1, W1c, W1h, W1e1, W3, and W5.

#### 3.5.1. Subclade W1

The HVS-I/HVS-II haplotype classified into subhaplogroup W1-G143A was found in seven individuals from the Serbian population, namely in individuals 13_Sb, 216_Sb [24], Ser_06 [51], Nish71 [5], and three individuals denoted as VP83 [53] (Appendix A). It is also recorded in two individuals from the neighboring countries of North Macedonia, and Bosnia and Herzegovina. Several branches (defined by transitions at nps 143, 263, 16,295 and 16,311 with 16,256) diverge from this node (Appendix A). Most haplotypes in these branches are differentiated by one or two mutation steps from this “Balkan specific” node. They are found in the Balkan, the Apennine, and the Iberian Peninsulas (Figure 7 and Appendix A).

Based on the complete mitogenome phylogeny, reconstructed using 210 sequences, we defined a new young subclade, W1j, defined by the transition at np 16,311 and dated to 1.8–3.07 kya, which groups haplotypes from Serbia (13_Sb and 216_Sb) and Italy (Appendix A).

#### 3.5.2. Subclade W1c

The HVS-I/HVS-II haplotype found in samples SS21 [47] and Nish43 [5] in the Serbian population was also found in six individuals from the Balkan Peninsula and one from Central Europe (Slovakia) (Appendix A). Another haplotype, differentiated by transitions at nps 185 and 16,325 from the previous, was detected in individual 229_Sb [24] (Appendix A). The haplotype found in sample Nish116 [5] is differentiated from its ancestral haplotype by a back mutation at np 16,223, and grouped with two samples from the Balkan Peninsula and one individual from the Caucasus region (Appendix A), which were all classified into subhaplogroup W1c. Subhaplogroup W1c has a diagnostic transition at the np 119 and is dispersed throughout Europe.

By phylogeny reconstruction based on 73 complete mitogenomes, we identified a new subclade, W1c2, defined by the transitions at nps 152 and 16,193. This subclade, which evolved around 3.26 kya, groups haplotypes found mostly in the Balkan Peninsula, including the 229_Sb (Appendix A).

#### 3.5.3. Subclade W1h and W1e1

Subhaplogroup W1h, defined by the transition at np 16,145, was represented in the Serbian population by the HVS-I/HVS-II haplotype found in sample Ser_05 [51], which is identical to that found in an individual from Slovenia (Appendix A). They form a branch characterized by transitions at nps 16,145 and 152, with haplotypes predominantly found in the Apennine and Balkan Peninsulas (Appendix A).

The haplotype found in sample Nish4 [5] belongs to the W1e1a branch, defined by the transition at np 16,324 from the ancestral HVS-I/HVS-II haplotype W1e1 which is characterized by the transition at np 16,295 (Appendix A). The W1e1a branch contains haplotypes found mostly in Central Europe and the South European peninsulas (Appendix A).

#### 3.5.4. Subclade W3

The HVS-I/HVS-II haplotype detected in the Serbian sample 257_Sb is classified into the subhaplogroup W3a and together with another haplotype from the Balkan Peninsula (Albanian from Vojvodina, [53]) creates a branch defined by the transitions at nps 333 and 16,176 (Appendix A). The ancestral haplotype for this branch was detected in 30 individuals of various geographical origins, including one ancient sample from Sardinia dated to the end of the 5th century BCE (MS10581, [56]). The HVS-I/HVS-II haplotype found in sample 24_Sb [9] was also detected in 20 individuals mostly originating from the Balkan Peninsula (Appendix A). Another Serbian haplotype, 183_Sb [24], differs from this predominantly Balkan node by the transitions at nps 199 and 263. Both haplotypes, 24_Sb and 183_Sb, were classified into subhaplogroup W3b2.

Phylogeny based on 56 complete mitogenomes allowed us to classify sample 257_Sb into the W3a1d subclade that emerged 5.77–9.63 kya (Appendix A). 24_Sb and 183_Sb haplotypes and three Bulgarian haplotypes form a new subclade, W3b2, defined by transitions at nps 210, 14,767, 16,172, and 16,231, with an estimated age of 2.71–4.61 kya (Appendix A). This W subclade may have originated in the Balkan Peninsula.

#### 3.5.5. Subclade W5

The HVS-I/HVS-II haplotype found in samples 27_Sb [9] and Brestovac11 (2 individuals) [5] was also detected among 34 individuals originating from different regions of Europe, Northern Africa, and the Arabian Peninsula (Appendix A). Several haplotypes were derived from this relatively frequent haplotype, including haplotype VP81 from the Serbian population [53], defined by the transition at np 16,093 (Appendix A).

Based on the phylogenetic analysis of 52 complete mitogenomes, sample 27_Sb belongs to the W5 subhaplogroup, which emerged 11.43–21.54 kya (Appendix A). According to available data on complete mitogenomes, 51 samples of various geographical affiliations are classified into this subhaplogroup (Appendix A).

### 3.6. Subhaplogroup X2

Subhaplogroup X2 is relatively rare, with the highest frequencies in the Near East among the Jewish, Druze, and Lebanese populations (Appendix A and Figure 6). In Europe, the highest prevalence of this haplogroup has been observed in the Balkan Peninsula, with the Romani population from Macedonia showing the highest frequency (Appendix A, Figure 6). The highest number of X2 haplogroup carriers can be found in the Near East, followed by the Apennine Peninsula and Central Europe (Appendix A). High HD values were detected in all analyzed regions, with the Balkan Peninsula showing the highest.

A haplotype network representing the subhalpogroup X2 was constructed using 458 HVS-I/HVS-II haplotypes detected in 969 individuals (Figure 8). In the Serbian population, subhaplogroup X2 is represented with nine haplotypes found in nine individuals (Figure 8 and Appendix A). These haplotypes were classified into the following subclades: X2, X2b+226, X2m’n, X2m, and X2n.

#### 3.6.1. Subclade X2

The X2 subclade gathers haplotypes of various geographical origins, including haplotypes found in three individuals from Serbia: 185_Sb [24], Nish 16, and Nish27 [5] (Appendix A). The HVS-I/HVS-II haplotype detected in individual 185_Sb, along with samples from Croatia and Bosnia and Herzegovina, belongs to a branch diverged from its ancestral haplotype by transition at np 16,294 (Appendix A). This ancestral haplotype has been identified in other populations from the Balkan Peninsula, the Apennine and Iberian Peninsulas, the Near East, and Central Asia. The haplotypes detected in individual Nish27 and one Bulgarian sample belong to the branch predominantly found in the Near East (Appendix A). The haplotype found in individual Nish16 differentiates from the ancestral haplotype present in the South European peninsulas, the Near East, and Central Asia due to transitions at nps 195, 16,264, and 16,376 (Appendix A).

Phylogeny based on 434 complete mitogenomes allowed us to classify sample 185_Sb into the X2q1a subclade previously described by Sarac et al. [22] (Appendix A). This subclade of possible Balkan origin emerged between 1.54 and 1.8 kya.

#### 3.6.2. Subclade X2b+226

The HVS-I/HVS-II haplotypes found in samples 36_Sb [24] and VP85 [53] are classified in the X2b subclade (Appendix A). The 36_Sb haplotype is widespread across the European continent and is ancestral to the one detected in sample VP85 (Appendix A). Sample VP85 belongs to the branch defined by the transversion at the np 16,184 which gathers haplotypes from Poland, Slovakia, and Russia (Appendix A).

Based on the analysis of complete mitogenomes, sample 36_Sb is classified in the X2b paragroup, defined by the transition 226 that evolved between 9.88 and 11.1 kya (Appendix A). An identical sequence was detected in the United Kingdom along with three samples of unknown origin.

#### 3.6.3. Subclades X2m’n, X2m, and X2n

Four HVS-I/HVS-II haplotypes observed in samples Nish58 [5], 137_Sb [24], VP86 [53], and Nish104 [5] were classified into subclades X2m’n, X2m1, and X2n, respectively (Appendix A). The haplotype found in sample Nish58 differentiates from the ancestral haplotype detected in Northern Europe by the transitions at nps 185, 188, and 16,376 (Appendix A). Haplotypes detected in samples 137_Sb and VP86 belong to the X2m1 branch together with samples from the Apennine, Balkan, and Iberian Peninsulas, and Central Europe (Appendix A). The haplotype of sample Nish104 belongs to the X2n subclade, which is defined by a transition at np 16,266 and the back mutation at np 263. It encompasses the haplotypes from the Balkan and Apennine Peninsulas, Central Europe (Poland and Slovakia), and the Caucasus (Georgia) (Appendix A).

Using complete mitogenomes to reconstruct the phylogeny of the X2m1 subclade, we identified a new branch, X2m1a, defined by a transition at np 16,192, which emerged between 7.88 and 9.22 kya (Appendix A). This branch gathers the Serbian haplotype and two haplotypes from the Apennine Peninsula (Appendix A).

## 4. Discussion

Phylogeographic analyses of the haplotypes belonging to the (sub)haplogroups seldom found in contemporary populations provide valuable insights into gene flow potentially linked to migrations from specific populations. Furthermore, for rare (sub)haplogroups, identifying geographic regions with increased levels of genetic diversity is particularly informative, as these areas may represent centers of origin for specific (sub)haplogroups, while populations inhabiting these regions during certain periods could be identified as source populations for those haplotypes. This information is crucial for inferring and understanding gene flow between populations.

Here, we discuss the prevalence, evolution, and origin of haplotypes belonging to rare subhaplogroups R0a, N1a, N1b, X2, I5, and W detected in the contemporary Serbian population. Subhaplogroups R0a, N1a, N1b, and X2 are characteristic for Near Eastern populations and Southwestern Asia [25,26,57] while I5 and W exhibit higher frequencies in European populations compared to the populations from the Near East [26,54]. All these rare (sub)haplogroups reflect the dynamic population history of the Balkan populations and numerous backward and forward migration events.

### 4.1. Subhaplogroup R0a

Both phylogeographic and founder analysis of subhaplogroup R0a, particularly its subclade R0a2r, suggest that its dispersal from the Arabian Peninsula to Southeastern Europe and the Mediterranean occurred by the end of the Pleistocene or during the early Holocene [25]. The HVS-I/HVS-II haplotype ancestral to the haplotype from the Serbian population is predominantly present in the Balkan and Arabian Peninsulas (Figure 2). Relatively high values of the genetic diversity parameters for R0a in the Balkan Peninsula (Appendix A) and the presence of haplotypes exclusively from the Balkan Peninsula in the newly defined R0a1a5 subclade (Appendix A) suggest that this subclade may have originated in the Balkans. The presence of an ancestral HVS-I/HVS-II haplotype in Southern Europe and the Arabian Peninsula and an estimated age of 1.54–1.8 kya for the R0a1a5 subclade suggest that this ancestral haplotype may have reached Southern Europe from the Near East during the Roman Empire’s control over the Mediterranean. This conclusion is further supported by the discovery of the R0a1a haplotype in the ancient remains from the Roman period’s necropolis of Viminacium in Serbia, dated between the 2nd and 3rd century CE (I15500, [20]).

### 4.2. Subhaplogroup N1a

The subhaplogroup N1a contains two major branches, European–Central Asian and African–South Asian, characterized by transversions 16,147A and 16,147G, respectively [52]. The 16,147A branch of N1a was widespread in the Central European Neolithic farmer populations [27] and is associated with the expansion of the first farmers into Central Europe [31,32,33,52].

The haplotypes belonging to the 16,147G branch of N1a were found in two Late Bronze Age individuals who lived near the Black Sea in the territory of modern-day Russia (RISE555 [13] and Kal1 [58]). These haplotypes are present in the modern populations of the Balkan Peninsula and Central and Eastern Europe as well (Figure 3). Complete genome analysis from ancient remains revealed that the Early Bronze Age migrations of the Yamnaya culture nomadic herders from the Pontic-Caspian steppe into Central and Northern Europe significantly impacted the formation of the genetic pool of contemporary European populations [13]. In the study by Haak et al. [15] the authors further demonstrated the importance of these migrations in shaping modern European genetic diversity and spreading Indo-European languages across Europe. Additionally, Yamnaya-related genetic components were identified in two Bronze Age individuals from the Balkan Peninsula (RISE595 and RISE596) [13]. Both branches of subhaplogroup N1a are present in Serbian and other Balkan populations, suggesting a possibility that these lineages arrived in the Balkan Peninsula in two instances: the 16,147A branch arrived with the Neolithic farmer populations during the Neolithic expansion and the 16,147G branch arrived with the nomadic herders, carriers of Yamnaya culture, during the Bronze Age migrations. In addition to these ancient migrations, the age of subclades of most likely Slavic origin, N1a1a2, N1a1a1a1b, and N1a1a1a1c (Appendix A), suggest their arrival in the Balkan region later in history, and most likely during the Migration Period.

### 4.3. Subhaplogroup N1b

Subhaplogroup N1b has the highest frequencies in Southwestern Asia while its occurrence in Europe is sporadic, mostly in the Central and Eastern Mediterranean [26]. Founder analysis of the HVS-I region variability suggests that this subhaplogroup spread through Europe during the Late Glacial period, before the Neolithic, thus making it possible that younger subclades like N1b2 evolved in situ in Europe [26]. Considering that HD parameter values for N1b are highest in the Near East, followed by the Balkans where high ND values were also observed (Appendix A), it could be possible that the carriers of subhaplogroup N1b arrived in Europe through the Balkan Peninsula during the Late Glacial period. N1b HVS-I/HVS-II haplotypes from the Serbian population were grouped with the other haplotypes from the Balkan Peninsula and haplotypes sporadically present in the Apennine Peninsula, Central, and Eastern Europe (Figure 4). Only the haplotype found in sample Nish33 was identical to those found outside the European continent (Figure 4). The majority of the haplotypes detected in the Serbian population belong to the subclades characteristic for the Balkans which could have evolved from the ancestral haplotypes after the migrations from the Near East around 15 kya, i.e., during the Late Glacial period. Reconstructed complete mitogenome phylogenies further support this idea since two samples from Serbia belong to the N1b1a7′8 (estimated age 8.44–15.04 kya) and younger N1b1a9 subclade (6.3–9.4 kya) which is predominantly found in Sardinia (Appendix A). A newly identified subclade N1b1a9a with an age estimate of 1.8–3.07 kya suggests the possible arrival of this subclade to the Balkan Peninsula from Sardinia during or after the Bronze Age.

### 4.4. Subhaplgroup I5

Although haplogroup I is the most common in Northern Europe, the highest values of the genetic diversity parameters were observed in the Near East and Southeastern Europe, indicating its Near Eastern origin [26,54]. Within the Serbian population, haplogroup I is represented by subhaplogroup I5 whose subclades I5a2a and I5b are specific for the Near East while I5a1 is European-specific [54]. Subhaplogroup I5 evolved around 18 kya in the Near East and spread throughout Europe around 10–11 kya [54]. Its carriers have been identified in Bronze Age populations of Southern [59] and Eastern Europe, as well as in the Caucasus [13]. They have also been observed in medieval populations of Central Europe [60]. The HVS-I/HVS-II haplotype of sample 173_Sb was grouped with haplotypes found in Southern Europe, mostly from the Balkans (Figure 5), suggesting that it likely evolved in the Balkan Peninsula. However, phylogeny reconstruction based on complete mitogenomes did not provide more details apart from classifying this haplotype into the I5a1 subclade (Appendix A). This highlights the need for more complete mitogenomic data from the Balkan Peninsula to better understand the evolution of this branch. The haplotype detected in sample 256_Sb was classified into the newly identified subclade I5d1 that most likely originated in Central Europe and spread into the Balkan Peninsula, presumably during the Migration Period.

### 4.5. Haplogroup W

The haplogroup W has a very complex evolutionary and migratory history. It exhibits the highest diversity of the HVS-I region in Southeastern Europe, northwestern Africa, and the Arabian Peninsula [26]. Based on the HVS-I haplotype network and the inclusion of Near Eastern lineages within primarily European subclades, [26] suggested that haplogroup W likely originated in Europe. On the other hand, Olivieri et al. [54] have suggested that this haplogroup evolved in the Near East during the Late Glacial, later rapidly spreading throughout Europe. Most W subclades, such as W3, W4, and W5, dispersed into Europe during the Late Glacial, while W1 expanded in the immediate postglacial period [54]. Some W lineages that had already been residing in Europe since the end of the ice age have expanded locally during the Neolithic period [26].

The analysis of the genetic diversity parameters indicates that Central Europe, the Apennine, and the Balkan Peninsulas represent centers of diversity for haplogroup W in Europe (Appendix A). HVS-I/HVS-II haplotypes found in the Serbian population are predominantly classified into the branches specific to Central Europe, the Balkan, and Apennine Peninsulas (branches in the subclades W1c, W1h, W1e1, and W3b). Phylogeny reconstruction allowed a more detailed classification of some Serbian haplotypes into several new subclades W1j, W1c2, and W3b2, or the already defined subclade W3a1d. Our analysis indicates that the origin of some of these subclades is in Central Europe (W3a1d) and the Balkans (W1c2, W3b2). Interestingly, the W1e1 subclade was discovered in ancient samples from medieval England and Finland and has a wide distribution in Europe although it is rarely found in Eastern Europe [61,62]. Unlike W1e1, haplotypes belonging to subclade W1c were found among Central European ancient samples from the Neolithic period [32,63].

Although results indicate that the majority of haplogroup W lineages from the Serbian population evolved in the Balkan Peninsula, we cannot exclude the possibility that some of the lineages (e.g., W3a1d) reached this region during the Bronze Age or Migration Period from Central and Eastern Europe. Additionally, subclade W3b, due to the high representation in the Near East, could have reached the Balkans from that region. As in the case of subclade I5, more available complete mitogenome sequences are necessary to accurately determine the origin of these lineages in the Serbian population.

### 4.6. Subhaplogroup X2

Haplogroup X displays a relict distribution across the Near East, reaching considerable frequencies and high diversity, suggesting that the Near East represents a contemporary refugium or reservoir of ancient diversity [64]. Subhaplogroup X2 is the most spread and frequent X subhaplogroup, although it has lower frequencies than more common haplogroups [26]. Our analysis estimated that subhaplogroup X2 evolved between 13.02 and 22.16 kya, which is aligned with the estimates of [26]. Some of the basal X2 branches are restricted to the Near East, the Caucasus, and Northern Africa, and the branch defined by the transition at np 225 includes Near Eastern, North African, and European-specific subclades, as well as the Native Americans’ X2a subclade [26].

The Balkan Peninsula has the highest X2 frequencies and values of the genetic diversity parameters in Europe, followed by other South European peninsulas (Appendix A). These results might reflect these regions’ role as glacial refugia, suggesting that the spread of some X2 subclades into Europe predates the Neolithic expansion. The origin of X2 haplotypes in the Serbian population is quite diverse and indicates different migratory events. Considering that the majority of haplotypes observed in the Serbian population belong to the subclades X2b, X2m’n, X2m, and X2n, which are also mostly found in Southern Europe, it is possible that some of these lineages evolved in the South European peninsulas, and that their ancestral haplotypes arrived in this region before or during the Neolithic expansion. X2m lineage found in the Serbian population may have originated in the Apennine or Balkan Peninsula while X2n might have evolved in the Balkan Peninsula. However, more complete mitogenome data for populations underrepresented in mtDNA studies is warranted to draw some reliable conclusions. Interestingly, the current distribution of HVS-I/HVS-II haplotype, ancestral to the sample VP85, suggests its Slavic origin and possible arrival in the Balkans in the Early Middle Ages during the Migration Period. Furthermore, both the HVS-I/HVS-II phylogeographic analysis and complete mitogenome phylogeny of X2q1 and its subclade X2q1a, observed exclusively in the South Slavic-speaking people of the Balkan Peninsula, demonstrate that this region was a place of local micro-differentiation processes, as previously reported by [22].

## 5. Limitations of the Study

Complete mitogenomes of contemporary and ancient humans, available nowadays in several databases such as GenBank [65], EMPOP (https://empop.online/, accessed 10 January 2025 [66]), Ian Logan webpage, and gnomAD [67], have inevitably contributed towards improving our understanding of the processes that have shaped the maternal landscape of the human populations. However, these high-resolution markers are still less abundant in comparison to HVS-I and HVS-II sequences, which are available for all the populations/regions here considered. Therefore, the trade-off due to the usage of lower resolution markers which are available for the populations/regions of interest indeed represents a limitation of our study. The same holds for the hypermutating sites frequently observed in heteroplasmic states which are present in HVS-I and HVS-II sequences and known to hamper the interpretation of phylogenies and haplotype networks. While further studies will provide complete mitogenomes from populations/regions of interest and thus improve our findings based on HVS-I and HVS-II sequences, hypermutating sites in HVS-I and HVS-II sequences prone to recurrent and back mutations have been omitted from both phylogenetic and phylogeographic analyses. Furthermore, whenever possible, we corroborated our inferences based on low-resolution markers by phylogeny reconstructions of rare mtDNA subhaplogroups using available complete mitogenome data. It is worth mentioning that although characterized by a lower resolution in comparison to complete mitogenomes, HVS-I, HVS-II, and HVS-III are still regarded as the standard genetic markers in forensic application and human identification (https://www.gednap.org, accessed 10 January 2025 [68,69]).

## 6. Conclusions

Phylogeographic and phylogeny analysis of rare mtDNA (sub)haplogroups performed in this study enabled us to gain new insights into how different migrations shaped the present-day mtDNA gene pool of the Serbian population. Our results add to the growing evidence pointing to the Balkan Peninsula as one of the glacial refugia from which the postglacial recolonization of Europe started. It also confirmed the Balkan Peninsula is an important center of mtDNA haplogroup diversity. Our data further corroborate previous findings that the mtDNA gene pool of the contemporary Serbian population was shaped by several ancient migration events, starting from the Neolithic through the Bronze Age and the Early Middle Ages to the present day.

## Figures and Tables

**Figure 1 genes-16-00106-f001:**
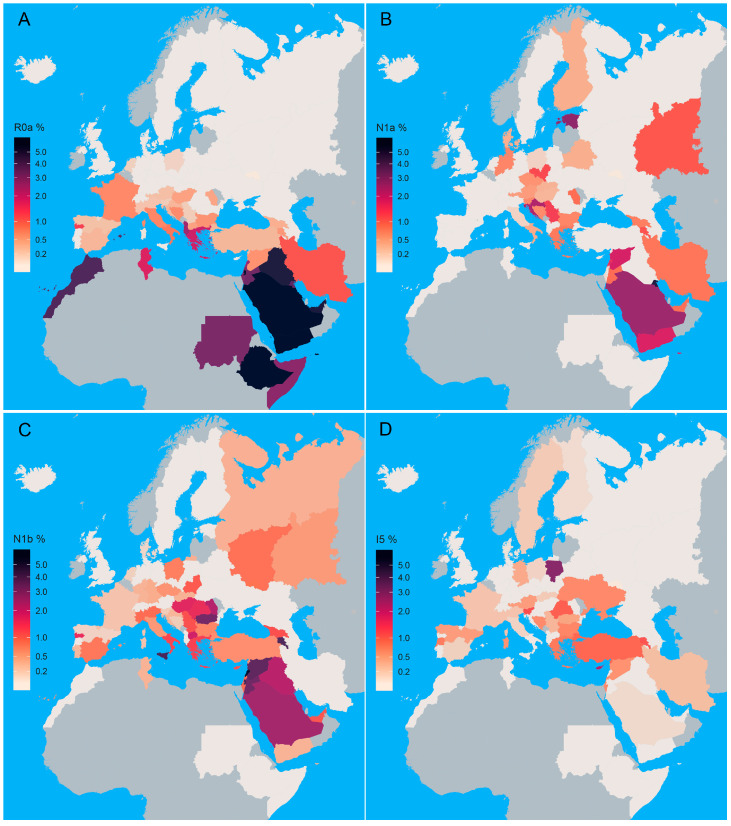
Choropleth map of R0a (**A**), N1a (**B**), N1b (**C**), and I5 (**D**) subhaplogroups’ frequency distribution. Frequency distribution in analyzed countries and regions is presented in Appendix A. The grey color represents the missing data for HVS-I/HVS-II haplotype distribution.

**Figure 2 genes-16-00106-f002:**
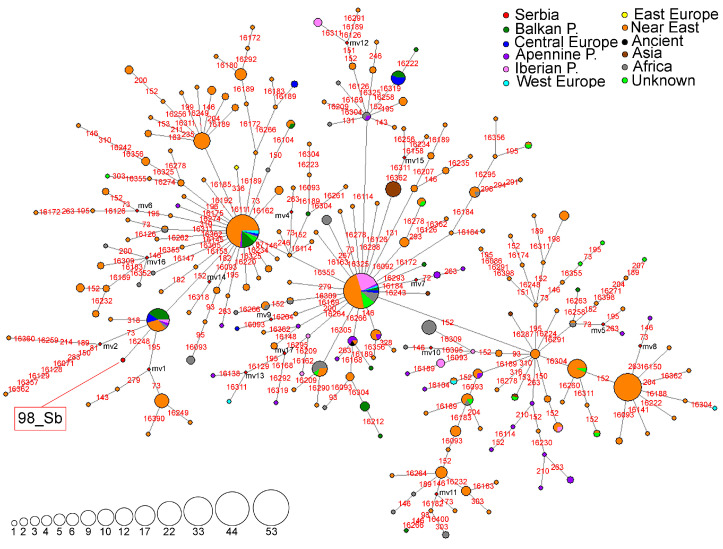
Phylogeographic network for R0a subhaplogroup based on 217 HVS-I/HVS-II haplotypes detected in 613 individuals. Differences from the Revised Cambridge Reference Sequence (rCRS) (NC_012920) are marked with red numbers representing nucleotide positions where the transitions occurred. Red rhomboids are the hypothetical haplotypes not detected in the analyzed sample (marked as mv1-17). The sizes of the circles are proportional to the number of detected haplotypes as depicted in the legend. The geographical origin of the analyzed samples is presented in the legend.

**Figure 3 genes-16-00106-f003:**
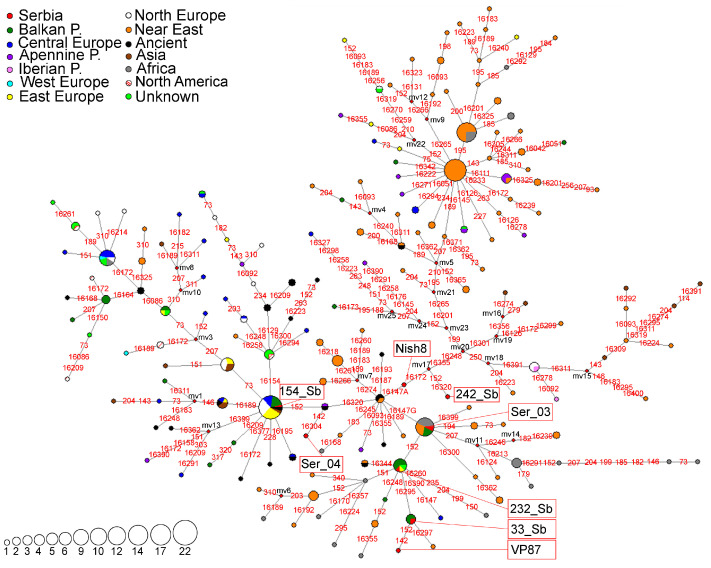
Phylogeographic network for N1a subhaplogroup based on 172 HVS-I/HVS-II haplotypes detected in 327 individuals, including 24 archaeological remains. Differences from rCRS are marked with red numbers representing nucleotide positions where the transitions occurred; transversions are marked with a red number and suffix representing the nucleotide change. Red rhomboids are the hypothetical haplotypes not detected in the analyzed sample (marked with mv1-25). The sizes of the circles are proportional to the number of detected haplotypes, as depicted in the legend. The geographical origin of the samples is presented in the legend.

**Figure 4 genes-16-00106-f004:**
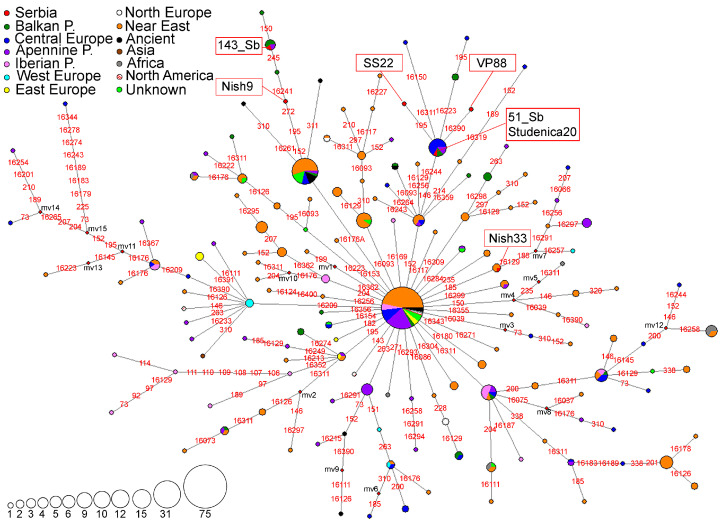
Phylogeographic network for N1b subhaplogroup based on 154 HVS-I/HVS-II haplotypes detected in 396 individuals, including 9 archaeological remains. Differences from rCRS are marked with red numbers representing nucleotide positions where the transitions occurred; transversions are marked with a red number and suffix representing the nucleotide change. Red rhomboids are the hypothetical haplotypes not detected in the analyzed sample (marked with mv1-15). The sizes of the circles are proportional to the number of detected haplotypes, as depicted in the legend. The geographical origin of the samples is presented in the legend.

**Figure 5 genes-16-00106-f005:**
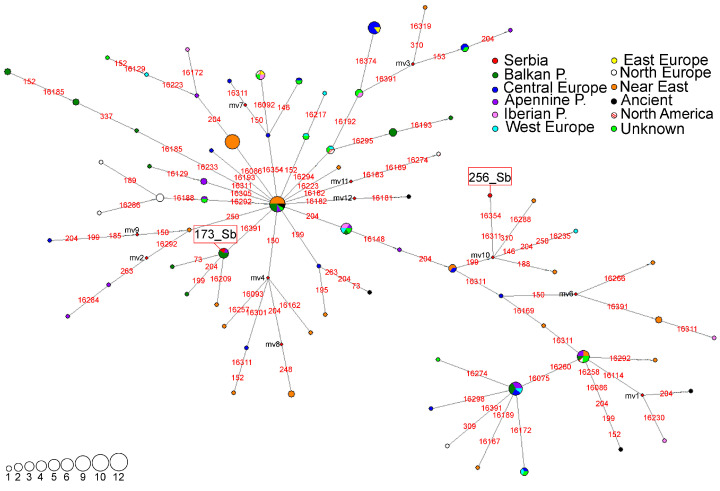
Phylogeographic network for I5 subhaplogroup based on 72 HVS-I/HVS-II haplotypes detected in 168 individuals, including 5 archaeological remains. Differences from rCRS are marked with red numbers representing nucleotide positions where the transitions occurred. Red rhomboids are the hypothetical haplotypes not detected in the analyzed sample (marked with mv1-12). The sizes of the circles are proportional to the number of detected haplotypes, as depicted in the legend. The geographical origin of the samples is presented in the legend.

**Figure 6 genes-16-00106-f006:**
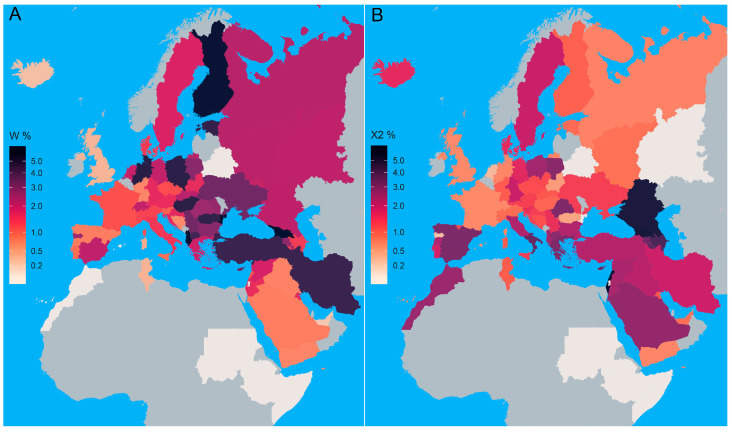
Choropleth map of W (**A**) and X2 (**B**) (sub)haplogroups’ frequency distribution. Frequency distribution in analyzed countries and regions is presented in Appendix A. The grey color represents the missing data for HVS-I/HVS-II haplotype distribution.

**Figure 7 genes-16-00106-f007:**
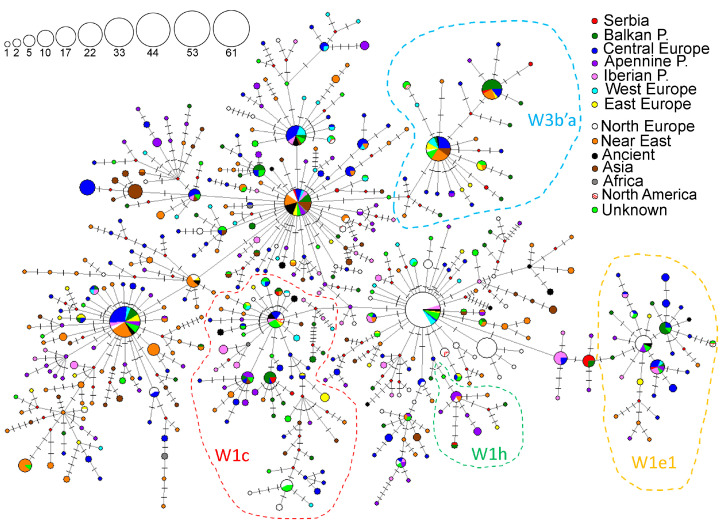
Phylogeographic network for W haplogroup based on 480 HVS-I/HVS-II haplotypes detected in 1026 individuals, including 32 archaeological remains. Differences from rCRS are marked with lines; red rhomboids are the hypothetical haplotypes not detected in the analyzed sample. The sizes of the circles are proportional to the number of detected haplotypes, as depicted in the legend. The geographical origin of the samples is presented in the legend.

**Figure 8 genes-16-00106-f008:**
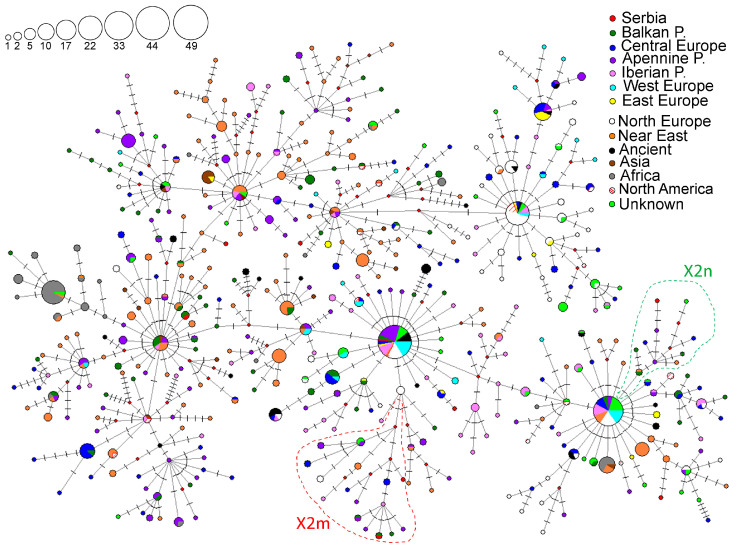
Phylogeographic network for the X2 subhaplogroup based on 458 HVS-I/HVS-II haplotypes detected in 969 individuals, including 41 archaeological remains. Differences from rCRS are marked with lines; red rhomboids are the hypothetical haplotypes not detected in the analyses. The sizes of the circles are proportional to the number of detected haplotypes, as depicted in the legend. The geographical origin of the samples is presented in the legend.

## Data Availability

All data with the accession numbers of publicly available sequences used in this study are available in the Appendix A.

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
