# Peer review of "Origin and Genealogy of Rare mtDNA Haplotypes Detected in the Serbian Population"

_genes, 2025, doi:10.3390/genes16010106_

Round 1
Reviewer 1 Report
Comments and Suggestions for Authors
The authors of study entitled “Origin and genealogy of rare mtDNA haplotypes detected in the Serbian population” aimed to explore the genetic diversity of human populations from Serbia and Balkans, through the analysis of rare mitochondrial DNA sub-lineages. The paper is well written and easily readable, the conclusions are quite supported by the results, but there are some points that must be improved. Hereafter my comments.
· Abstract: I think that authors should make explicit that the dataset was entirely based on previously published sequences and specify if they used modern or ancient DNA data, or both.
· Since the manuscript focuses on rare mtDNA sub-lineages and the HG classification is a crucial point to present and discuss the results, I think that authors should use Haplogrep 3 instead of Haplogrep 2. There are some differences between the two versions; I know that this is a great effort, but I suggest checking and verifying any inconsistency, especially at a sub-haplogroup level.
· Figure 1: the legend is not so readable. Please use capital letters to indicate each map (i.e. A for R0a, B for N1a, etc) and make it explicit also in the legend.
· I think that authors should indicate the number of sequences for each sub-lineage also in the text and not only in the figure legends. The entire manuscript is based on many published data; thus, it is important to specify the total number of sequences (for each sub-hg analysis) and how many of them are ancient and modern.
Despite all the analyses here performed, my last comment is about your choice of submitting this manuscript as an article rather than a review. Nevertheless, if authors check the HG classification with Haplogrep3, verify any possible inconsistency e provide a table reporting the output of both Haplogrep versions, I think that this manuscript could be considered for the publication as an article.
Thanks for your efforts.
Author Response
Reviewer 1
The authors of study entitled “Origin and genealogy of rare mtDNA haplotypes detected in the Serbian population” aimed to explore the genetic diversity of human populations from Serbia and Balkans, through the analysis of rare mitochondrial DNA sub-lineages. The paper is well written and easily readable, the conclusions are quite supported by the results, but there are some points that must be improved. Hereafter my comments.
Thank you for your kind suggestions for the improvement of our manuscript which we have implemented in the revised version of our manuscript which we provide.
- Abstract: I think that authors should make explicit that the dataset was entirely based on previously published sequences and specify if they used modern or ancient DNA data, or both.
- Thank you for the observation. Accordingly, in the abstract, we now provide additional information regarding the dataset used in the study (publically available data from both contemporary and ancient DNA samples) (lines 28-30) to describe our dataset more precisely, as suggested.
- Since the manuscript focuses on rare mtDNA sub-lineages and the HG classification is a crucial point to present and discuss the results, I think that authors should use Haplogrep 3 instead of Haplogrep 2. There are some differences between the two versions; I know that this is a great effort, but I suggest checking and verifying any inconsistency, especially at a sub-haplogroup level.
- Thank you for the suggestion. Yes, we are fully aware of the fact that Haplogrep 2 and 3 can yield somewhat different classifications, especially at a sub-haplogroup level because, in fact, we did carry out classification using Haplogrep 2 and updated version 2.4, and based on the suggestion of Reviewer 1, with Haplogrep 3. We found certain discrepancies in classifications based on Haplogrep 2 and updated version 2.4, and in some cases between classifications based on Haplogrep 2.4 and 3. We used Haplogrep when haplogroup classification was not performed in the original publication. Furthermore, we observed at several instances that classifications given in original publications deviate from those obtained by using Haplogrep 2, 2.4 and 3. To overcome these obstacles, and to provide all the information to the readers, we improved our Supplementary tables S3-S8 by adding the following columns: original classification, as given in the original publication or the classification as listed on the Ian Logan public database of the available published complete mitogenome sequences (http://www.ianlogan.co.uk/sequences_by_group/haplogroup_select.htm), classification based on Haplogrep 2.4, and classification based on Haplogrep 3. We also highlight several instances when discrepancies between classifications based on complete mitogenome data, given in original publications, do not coincide with Haplogrep classifications, based on lower-resolution HVSI-HVSII sequences.
Figure 1: the legend is not so readable. Please use capital letters to indicate each map (i.e. A for R0a, B for N1a, etc) and make it explicit also in the legend.
- Thank you for your input, the Figures have now been updated per your suggestion.
- I think that authors should indicate the number of sequences for each sub-lineage also in the text and not only in the figure legends. The entire manuscript is based on many published data; thus, it is important to specify the total number of sequences (for each sub-hg analysis) and how many of them are ancient and modern.
- Thank you for your suggestion. Although we presented the number of sequences for each sub-lineage only in figure legends because we did not want to burden the main text with too many numbers, we agree that it would be beneficial for the readers to have this information in the main text as well. Therefore, we have now included this information in the main text, as you suggested (lines: 181-182, 187, 211-212, 221, 232, 248-249, 258, 275, 289-290, 300, 305, 331-332, 349-350, 355, 369, 394, 405, 417-418, 434, 439).
Despite all the analyses here performed, my last comment is about your choice of submitting this manuscript as an article rather than a review. Nevertheless, if authors check the HG classification with Haplogrep3, verify any possible inconsistency e provide a table reporting the output of both Haplogrep versions, I think that this manuscript could be considered for the publication as an article.
- Thank you for your insight. This is something that we as the authors have already discussed. However, since we performed several analyses based on the collected publicly available data, we believe that our work can be categorized as a meta-analysis rather than a review which typically just simply collects the data and reviews the current knowledge. More precisely, we have:
- reassessed the frequencies of analysed rare haplogroups in different populations and geographical regions
- assessed the genetic diversity indices for analysed haplogroups in different geographical regions
- constructed new haplotype networks using HVS-I/HVS-II data and phylogenies based on complete mitogenome sequences
- and also identified several new subclades and evaluated the age of several subhaplogroups.
Furthermore, per your suggestion, we have conducted haplogrouping using both Haplogrep 2 and 3 software.
Thanks for your efforts.
Reviewer 2 Report
Comments and Suggestions for Authors
The paper aims to delineate the origin and genealogy of rare mtDNA haplotypes in the Serbian population. I have the following remarks:
1. The authors claim that they obtained the genetic data of their study from available published data and refer to Table S1. Here, not % values but instead individuals belonging to the rare haplogroups should be shown.
2. The authors work in this study with HVS-I/HVS-II data only. Among these sites are hypermutating sites frequently observed in heteroplasmic state (e.g. 189, 310). The presented trees are therefore highly error prone. It would be much more convincing if the constructed trees would have been verified with coding region mutations. The authors should therefore show a tree on the basis of the mentioned 1,426 complete mitogenomes and compare it with the HVS-I/HVS-II tree to convince the critical reader. Furthermore, in a statement about the limitations of this study the remutation/heteroplasmy problem (regarding HVS-I/HVS-II trees) should be mentioned.
3. For the identified novel subclades also the relevant coding region mutations should be delineated.
4. In gnomAD 56,434 whole genome data (which include complete mitogenomes and info of ethnic origin) are deposited. Did the authors try to use this additional resource?
Author Response
Reviewer 2
The paper aims to delineate the origin and genealogy of rare mtDNA haplotypes in the Serbian population.
- Thank you for your valuable suggestions for the improvement of our manuscript, which we have addressed and implemented accordingly in the revised version of our manuscript which we provide.
I have the following remarks:
- The authors claim that they obtained the genetic data of their study from available published data and refer to Table S1. Here, not % values but instead individuals belonging to the rare haplogroups should be shown.
- Thank you for your valuable comment. As suggested, we have updated Table S1, which now contains the actual number of individuals in the analysed populations for the appropriate mtDNA haplogroups as well as % values.
- The authors work in this study with HVS-I/HVS-II data only. Among these sites are hypermutating sites frequently observed in heteroplasmic state (e.g. 189, 310). The presented trees are therefore highly error prone. It would be much more convincing if the constructed trees would have been verified with coding region mutations. The authors should therefore show a tree on the basis of the mentioned 1,426 complete mitogenomes and compare it with the HVS-I/HVS-II tree to convince the critical reader. Furthermore, in a statement about the limitations of this study the remutation/heteroplasmy problem (regarding HVS-I/HVS-II trees) should be mentioned.
- Thank you for the comment. We are fully aware of the fact that usage of HVSI-HVSII sequences together with diagnostic mutations in coding regions would yield trees with less errors. Unfortunately, the majority of the published data for different populations we used for the analysis consists only of HVS-I/HVS-II sequences, i.e. did not have information on haplogroup diagnostic polymorphisms from the coding region. In addition, the available data are rather heterogeneous, because they comprise: HVS-I/HVS-II sequences of different lengths, additional HVS-III region, and some but not all coding region polymorphisms. Therefore, we opted for the least common denominator for all of the published data, which are the sequence lengths from positions 72-340 and 16024-16400 which we stated in the section Material and Methods. Because of this limitation, we also reconstructed phylogenies using the complete mitogenome sequences reported in the supplementary data (Supplementary figures S1-S21). Creating haplotype networks using complete mitogenomes thus seems redundant since we already used complete mitogenomes to reconstruct phylogenies that we compared with the haplotype networks based on HVS-I/HVS-II polymorphisms. In order to address these limitations, we provided a new section in our manuscript - section 5. Limitations of the study (lines 625-645):
“Complete mitogenomes of contemporary and ancient humans, available nowadays in several databases such as GenBank [65], EMPOP (https://empop.online/ [66]), Ian Logan webpage, and gnomAD [67], have inevitably contributed towards improving our understanding of the processes that have shaped the maternal landscape of the human population. However, these high-resolution markers are still less abundant in comparison to HVS-I and HVS-II sequences, which are available for all the populations/regions that were the focus of our study. Therefore, the trade-off due to the usage of lower resolution markers which are available for the populations/regions of interest indeed represents a limitation of our study. The same holds for the hypermutating sites frequently observed in heteroplasmic states which are present in HVS-I and HVS-II sequences and known to hamper the interpretation of phylogenies and haplotype networks. While further studies will provide complete mitogenomes for populations/regions of interest and thus improve our findings based on HVS-I and HVS-II sequences, hypermutating sites in HVS-I and HVS-II sequences prone to recurrent and back mutations have been omitted from both phylogenetic and phylogeographic analyses. Furthermore, whenever possible, we corroborated our inferences based on low-resolution markers by phylogeny reconstructions of rare mtDNA subhaplogroups using available complete mitogenome data. It is worth mentioning that although characterized by a lower resolution in comparison to complete mitogenomes, HVS-I, HVS-II, and HVS-III are still regarded as the standard genetic markers in forensic application and human identification.”.
- For the identified novel subclades also the relevant coding region mutations should be delineated.
- Thank you for noticing that this important information has not been provided. Although we have taken into account coding region mutations in the phylogeny reconstruction and delineation of novel proposed subclades in MtPhyl software, we accidentally omitted to mention the defining mutations in some instances in the main text. Therefore, in the revised version of the manuscript, we have included coding and control region mutations wherever they were left out (Lines: 274-276, 274-276, 311-312, 350, 396-397, 465).
- In gnomAD 56,434 whole genome data (which include complete mitogenomes and info of ethnic origin) are deposited. Did the authors try to use this additional resource?
- Thank you for pointing to the mentioned database with a rather large number of whole genome data including complete mitogenomes. We, however, didn’t use complete mitogenomes available in this database but rather those available in the Ian Logan database of complete mitogenomes which is regularly being updated (http://www.ianlogan.co.uk/sequences_by_group/haplogroup_select.htm). Nonetheless, we do mention this database in the revised version of our manuscript as a potential resource of complete mitogenomes which can contribute towards the improvement of our understanding of the processes that have shaped the maternal landscape of the human population (lines 626-629).
Round 2
Reviewer 1 Report
Comments and Suggestions for Authors
The manuscript “Origin and genealogy of rare mtDNA haplotypes detected in the Serbian population” was improved following reviewers’ suggestions. There are only few minor points to be addressed.
Line 29: please replace “included already published of 3,499 HVS-I/HVS-II sequences and 1,426 complete mitogenomes from” with “included already published sequences from 3,499 HVS-I/HVS-II and 1,426 complete mitogenomes belonging to”
Line 629-630: please modify with “human populations”
Line 631-632: please replace “that were the focus of our study” with “here considered”
Line 637: please replace “complete mitogenomes for populations/regions” with “complete mitogenomes from populations/regions”
Line 642-645: “It is worth mentioning that although characterized by a lower resolution in comparison to complete mitogenomes, HVS-I, HVS-II, and HVS-III are still regarded as the standard genetic markers in forensic application and human identification” please provide some relevant and recent references.
Author Response
Thank you for your effort and corrections. Your input has considerably improved our manuscript. We have implemented all of your corrections in the manuscript. As you suggested, we added more references concerning the usage of HVS-I/HVS-II and HVS-III regions in contemporary forensics laboratories.
Reviewer 2 Report
Comments and Suggestions for Authors
The authors have addressed all of my concerns.
Author Response
Thank you for all of your previous comments and suggestions, they helped us improve the quality of our manuscript.